# Mechanistic Study on the Possibility of Converting Dissociated Oxygen into Formic Acid on χ-Fe_5_C_2_(510) for Resource Recovery in Fischer–Tropsch Synthesis

**DOI:** 10.3390/molecules28248117

**Published:** 2023-12-15

**Authors:** Ning Ai, Changyi Lai, Wanpeng Hu, Qining Wang, Jie Ren

**Affiliations:** 1College of Chemical Engineering, Zhejiang University of Technology, Hangzhou 310014, China; aining@tsinghua.org.cn (N.A.); laicy13755386760@163.com (C.L.); 2College of Biological, Chemical Sciences and Engineering, Jiaxing University, Jiaxing 314001, China; hu688@zjxu.edu.cn; 3National Demonstration Center for Experimental Chemistry and Chemical Engineering Education, Zhejiang University of Technology, Hangzhou 310014, China; wqn989@zjut.edu.cn; 4Key Laboratory for Green Chemical Technology of Ministry of Education, R&D Center for Petrochemical Technology, Tianjin University, Tianjin 300072, China

**Keywords:** χ-Fe_5_C_2_, Fischer–Tropsch synthesis, hydrogen coverage effect, formic acid, dissociated oxygen

## Abstract

During Fischer–Tropsch synthesis, O atoms are dissociated on the surface of Fe-based catalysts. However, most of the dissociated O would be removed as H_2_O or CO_2_, which results in a low atom economy. Hence, a comprehensive study of the O removal pathway as formic acid has been investigated using the combination of density functional theory (DFT) and kinetic Monte Carlo (kMC) to improve the economics of Fischer–Tropsch synthesis on Fe-based catalysts. The results show that the optimal pathway for the removal of dissociated O as formic acid is the OH pathway, of which the effective barrier energy (0.936 eV) is close to that of the CO activation pathway (0.730 eV), meaning that the removal of dissociated O as formic acid is possible. The main factor in an inability to form formic acid is the competition between the formic acid formation pathway and other oxygenated compound formation pathways (H_2_O, CO_2_, methanol-formaldehyde); the details are as follows: 1. If the CO is hydrogenated first, then the subsequent reaction would be impossible due to its high effective Gibbs barrier energy. 2. If CO reacts first with O to become CO_2_, it is difficult for it to be hydrogenated further to become HCOOH because of the low adsorption energy of CO_2_. 3. When the CO + OH pathway is considered, OH would react easily with H atoms to form H_2_O due to the hydrogen coverage effect. Finally, the removal of dissociated O to formic acid is proposed via improving the catalyst to increase the CO_2_ adsorption energy or CO coverage.

## 1. Introduction

Fischer–Tropsch synthesis, which is able to convert synthesis gas into hydrocarbons (alkane and olefin) and oxygenated organic compounds (higher alcohols) [1,2,3], could develop the diversification of the world’s fuel supply [4] and the possibility of cleaner fuels free of sulfur, nitrogen, and aromatic compounds [5,6,7]. Fischer–Tropsch synthesis is a very complex multiphase catalytic process [8], with the shortcomings of wide product distribution, low target selectivity, and poor catalyst stability [9,10]. However, improving the economics of Fischer–Tropsch synthesis requires not only better selectivity [4] but also higher atom utilization. A literature review revealed that O atoms are dissociated and left on the catalyst surface during Fischer–Tropsch synthesis [11,12,13,14,15], and the majority of these dissociated O are removed from the surface in the form of H_2_O or CO_2_ [16,17,18,19], which would reduce the atom economy. There are two ways to solve this problem: one is to promote the further reaction of oxygen-containing compounds in the product to form valuable products, such as the hydrogenation of CO_2_ to hydrocarbons [20]; the other is to find a product with high economic benefits that can replace CO_2_ and H_2_O. The former improves the economic benefit to a certain extent. However, it does not solve the problem of using the oxygen atoms, so it is better to find a substitute.

Theoretically, there are three main ways to remove dissociated O in Fischer–Tropsch synthesis (as shown in Figure 1): 1. Dissociated O would react with hydrogen to be removed as H_2_O (a by-product with no economic value) [12,14]; 2. dissociated O would react with CO to be removed as CO_2_ (a by-product with no economic value and an additional greenhouse effect) [14,21]; and 3. dissociated O would react with hydrogen and CO to be removed as formic acid or polyacid. It is worth noting that when dissociated O is removed as alcohol [22,23,24], though it appears to produce an economically valuable product, the dissociated O that produces the alcohol is derived from CO, and essentially no additional dissociated O is removed, Therefore, it is necessary to form a product in which one C atom could take away two or more O atoms; the products that meet this condition are formic acid and polyacids. In particular, formic acid, besides being nontoxic, having a high energy density, and being renewable and degradable, has great potential in green organic synthesis and biomass conversion [25] and has wide applications in new energy utilization [26,27]. However, formic acid has not been found in Fischer–Tropsch synthesis in the literature, which could be attributed to unsatisfied catalysts or the failure of the reaction itself. Therefore, to investigate whether formic acid can theoretically be formed or not and to analyze the main factors limiting the formation of formic acid on the catalyst surface, the O removal pathway from the catalyst surface must be discussed.

At present, the O removal pathways from the surface of Fischer–Tropsch synthesis catalysts are not frequently reported, and studies have mainly investigated the removal mechanism of dissociated O in the form of H_2_O or CO_2_. For example, the mechanism of O removal on Fe(710) has been completed [21]; the following aspects were considered: the direct addition and disproportionation reactions of OH with OH to produce H_2_O; the direct reaction of dissociated O with CO, and the dehydrogenation of COOH to produce CO_2_. Analyzing the apparent Gibbs barrier energy, the results show that the removal of dissociated O from on the Fe(710) surface favors the production of H_2_O rather than CO_2_. In addition, it was shown that the hydrogen coverage effect has an impact on the O removal pathway [14]. At low hydrogen coverage, carbon dioxide is more likely to be produced through the decomposition of CO and OH, and H_2_O is produced through the pathway of two OH. At high hydrogen coverage, CO_2_ is more likely to be formed via the CO + O → CO_2_, and H_2_O is formed via the OH + H → H_2_O. The results of this study show that CO_2_ is more likely to be formed via the decomposition of CO + OH → COOH. However, it is inconclusive whether dissociated oxygen can be removed from the catalyst surface in the form of formic acid. 

Fe-based comprises one of the representatives of traditional Fischer–Tropsch synthesis catalysts, due to its cost, wide operating temperature, and high activity [28,29]. The χ-Fe_5_C_2_ phase is considered to be the main active phase of Fe-based Fischer–Tropsch synthesis catalysts [30], and the χ-Fe_5_C_2_(510) surface is the main exposed surface of this carbide phase [31], so, in this paper, the χ-Fe_5_C_2_(510) surface is chosen as the represent facet to investigate the O removal pathway and reaction mechanism in Fischer–Tropsch synthesis on the χ-Fe_5_C_2_ (510) surface. Therefore, whether the Fischer–Tropsch synthesis itself is conducive to the formation of formic acid products is considered first, through examining the major reaction pathways for formic acid formation. Then, representative and relatively simple oxygen-containing products (carbon dioxide, hydrogen, formaldehyde, and methanol) are selected, and the competing relationships of formic acid to their formation processes are discussed to explain the failure to remove dissociated O as formic acid.

In this work, the elementary reactions of the Fischer–Tropsch synthesis O removal pathway on the surface of χ-Fe_5_C_2_(510) are first considered as comprehensively as possible using density functional theory (DFT) calculations. The basic data obtained via DFT calculations are applied to kinetic Monte Carlo (kMC) simulations, and the reaction frequencies of various elementary reactions are obtained. The major reaction pathways and the rate control steps when the O atom are removed as CO_2_, H_2_O, formaldehyde, methanol, and formic acid are clarified, according to the form in which the oxygen atoms are removed; also, the main factors limiting the dissociated O removed as formic acid are determined. This study could provide a theoretical basis and direction for the simultaneous production of Fischer–Tropsch synthesis products, using carbon resources in CO and formic acid, using dissociated oxygen resources in Fischer–Tropsch synthesis. 

## 2. Results and Discussion

### 2.1. Computational Models and Methods

#### 2.1.1. DFT Calculation Method

In this work, the effect of the magnetic properties of the χ-Fe_5_C_2_ phase on the energy and structure of the system is considered using the Vienna Ab initio Simulation Package (VASP) based on the periodic flat plate model and the spin polarization DFT method [32,33,34,35]. The exchange correlation energy is described via the projective affixed plane wave (PAW) method using the Perdew–Burke–Ernzerhof (PBE) function of the generalized gradient approximation (GGA) [33,36]. The plane wave cut-off energy is set to 400 eV [37]. The Brillouin zone integrals calculated via the bulk phase model and the surface model were modeled with 3×5×5 and 2×2×1 grids, respectively [38]. The electron occupancy state is determined using the Methfessel–Paxton method, where the smearing parameter is set to 0.2 eV [39]. The Jonson (BJ) damped DFT-D3 method is used to correct for van der Waals effects [40,41].

The transition state search is implemented using the climbing image nudged elastic band (CI-NEB) method [42,43]. The accuracy of the DFT calculations is set as follows: the convergence criteria for the self-consistent calculation of the electrons (SCF), the geometrically optimized calculation, and the transition state calculation are 1.0 × 10^−6^ eV, 0.03 eV/Å and 0.05 eV/Å, respectively. After the convergence of the transition state calculation, the transition state structure is subjected to frequency analysis and verified, and if only one imaginary frequency exists in the calculated structure, then the structure passes the test. The adsorption energy *E_ads_* (eV) is calculated using Equation (1):(1)$$E_{ads}=E_{\left(adsorbate/slab\right)}-[E_{\left(slab\right)}+E_{\left(adsorbate\right)}]\;$$
where $E_{\left(adsorbate/slab\right)}$ (eV), $E_{\left(slab\right)}$ (eV), and $E_{\left(adsorbate\right)}$ (eV) represent the total energy of the adsorption system, the surface energy of the unadsorbed mass, and the energy of the adsorbent itself, respectively.

The effects of the zero point energy (ZPE) correction, the standard molar vibrational internal energy contribution, and the standard molar vibrational entropy were considered to better evaluate the standard molar Gibbs free energy for each species. The standard molar Gibbs free energy ($G^{0}$) for each species was calculated using Equation (2), as follows:(2)G0=E+zPE+U0+γRT(1+lnpp0)−TS0

Using the standard molar Gibbs free energy ($G^{0}$) calculated using the above equation, insert it into the Gibbs reaction energy and Gibbs free energy formulae G_r_ = G(FS) − G(IS) and G_a_ = G(TS) − G(IS), respectively, where G(IS) represents the initial Gibbs energy (IS), G(FS) represents the final Gibbs energy (FS), and G(TS) represents the transition Gibbs energy (TS) for the final Gibbs energy (FS), and G(TS) for the transition Gibbs energy (TS).

As the reaction network studied in this work is complex, with a large number of concatenated and parallel reactions, the coverage of the intermediates in the concatenated reactions is an important parameter affecting the reaction rates according to the mass low. Therefore, when searching for the optimal reaction path, it is not sufficient to determine whether an elementary reaction is a tachyonic step or not simply by characterizing its reaction rate in terms of its Gibbs barrier energy. In this paper, the effective Gibbs barrier energy $G_{a,eff}^{k}$ is used to account for the effect of the coverage of the reactants on the catalyst surface on the rate of formation of different products [44,45]. The effective Gibbs barrier energy $G_{a,eff}^{k}$ (eV) for the kth reaction step in a given cascade is given by Equation (3):(3)$$G_{a,eff}^{k}=G_{a}^{k}+\sum_{i=j}^{k-1}G_{r}^{i}$$
where $G_{a}^{k}$ (eV) and $G_{r}^{i}$ (eV) are the Gibbs barrier energy of the *k*^th^ step and the heat of reaction of the *i*^th^ step, respectively. *j* is the *j*^th^ reaction step, which is the primitive step with the lowest product energy in the reaction chain. 

#### 2.1.2. DFT Calculation Model

In this paper, the bulk phase χ-Fe_5_C_2_ model is optimized as a = 11.570Å, b = 4.501Å, c = 4.986Å, and β = 97.56°, which is consistent with the experimental cell parameters of a = 11.588Å, b = 4.579Å, c = 5.059Å, and β = 97.95° [46]; therefore, the calculation results are reliable. Based on the optimized bulk phase χ-Fe_5_C_2_ model, a χ-Fe_5_C_2_(510) surface model with two layers of Fe atoms and four layers of C atoms was developed. The lower one layer of Fe atoms and two layers of C atoms were fixed, while the other atoms and adsorbates were allowed to relax. The vacuum layer was set to 13 Å. And the geometry optimization and transition state searches for adsorbed species on the χ-Fe_5_C_2_(510) surface were performed on the p(2 × 1) supercell surface.

According to previous studies [14], hydrogen coverage has a great influence on the Fischer–Tropsch synthesis mechanism. The results show that on the surface of clean χ-Fe_5_C_2_(510), H_2_ dissociates to produce H atoms adsorbed at the triplet position first and accumulates on the surface to form hydrogen-covered χ-Fe_5_C_2_(510) facets. Then, on the hydrogen-covered χ-Fe_5_C_2_(510), H_2_ dissociates to form bridge-site H atoms with strong hydrogenation radical reaction activity, which are the main H atoms involved in the surface hydrogenation radical reaction. The consideration of hydrogen-covering effects can make the simulated catalyst surface more realistic. Hence, one layer of inlet hydrogen is also considered in the modelling. Top and side views of the investigated H-covered χ-Fe_5_C_2_(510) surface are shown in Figure 2.

#### 2.1.3. Models and Methods for Calculating kMC

The lattice gas model (LGM), with the variable step size method (VSSM), was used for the kMC simulations in this work [47,48]. A number of processes, adsorption, dissociation, migration, and reaction, as well as the associated species, were defined based on the DFT calculations. For the adsorption process, the adsorption rate *k_ads_* (1/s) [49] of the reactants is described by Equation (4):(4)$$k_{ads}=\frac{PA_{site\sigma }}{\sqrt{2\pi mk_{B}T}}$$
where *σ* is the dimensionless adhesion coefficient, the value of which is often taken as 1 [50]. *p* (Pa) and *T* (K) are the partial pressure and absolute temperature of CO under the simulated conditions, *A_site_* (m^2^) is the area of the adsorption site of a single molecule, and *m* (kg) is the molecular mass; *k_B_* (J/K) is the Boltzmann constant. The rate constants *k_rec_* of activated processes were all calculated according to the transition state theory by Equation (5): (5)$$k_{i,j}\left(T\right)=\frac{q_{TS}^{vib}}{q_{i}^{vib}}\frac{k_{B}T}{h}e^{-\frac{∆E_{ij}}{k_{B}T}}$$
where *TS* stands for the transition state joining the initial state *i* with the final state *j*, *T* (K) is the absolute temperature, *k_B_* (J/K) is defined as above, and *h* (J*s) is the Planck constant. $q_{TS}^{vib}$ and $q_{i}^{vib}\;$ are the vibrational partition functions for the *TS* and i states, respectively, the latter being approximated from the vibrational frequencies of the respective state using the harmonic approximation. $∆E_{ij}$ stands for the activation energy barrier of the process obtained through DFT calculations.

The lattice gas model is the model used for the kMC simulations in this paper. The calculations were performed using a 64 × 64 two-dimensional periodic model, with each lattice representing a p(1 × 1) χ-Fe_5_C_2_(510) surface. A total of 32 adsorption sites are set up within the lattice, of which 10 sites represent the top sites for Fe atoms corresponding to the p(1 × 1) χ-Fe_5_C_2_(510) surface atoms one at a time. Another 10 sites represent triple-adsorption sites of Fe atoms, 9 sites represent bridge sites of Fe atoms, and 3 sites represent quadruple-adsorption sites of Fe atoms, of which 2 quadruple-adsorption sites are top sites for C atoms, and 1 quadruple adsorption site is empty. Figure 3 shows the kMC model used for the calculation and the corresponding DFT model.

### 2.2. The Design of Dissociated O Removal Pathways on Hydrogen-Covered χ-Fe_5_C_2_(510) Surface

To achieve the objectives of this paper, a possible dissociated O removal reaction network is designed for the generation of water, carbon dioxide, formic acid, and methanol species on hydrogen-covered χ-Fe_5_C_2_(510) surfaces with dissociated O, bridge site H, top site CO, and OH as feedstock, as shown in Figure 4.

The ideas for the designed possible reaction network is as follows: firstly, for the dissociated O removal pathway in the form of H_2_O and CO_2_, the two-step hydrogenation of dissociated O to produce H_2_O, as well as the direct reaction of CO with dissociated O to produce CO_2_, are considered, mainly based on the research results of Zhang et al. [14]. Secondly, the reaction pathway for the stepwise hydrogenation of CO to produce methanol is considered, and contains the following two parallel reactions: 1. hydrogenation reaction at the C site; 2. hydrogenation reaction at the O site, which is the main reaction pathway for the black channel in Figure 4. Finally, finding the possible surface species that can react with dissociated O or OH in the main pathway to form formic acid are considered and denoted as the red O pathway or the blue OH pathway. 

### 2.3. The Formic Acid Formation Pathway on Hydrogen-Covered χ-Fe_5_C_2_(510) Surface

According to the designed reaction pathway in Figure 4, there are eight possible reaction pathways for the formation of formic acid on χ-Fe_5_C_2_(510). They could be divided into three categories for discussion, as shown in Table 1, which are: 1, CO pathway (pathways 1, 2, 3, and 4); 2, OH pathway (pathways 5 and 6); 3, CO_2_ pathway (pathways 7 and 8). Table 1, below, gives a detailed description of the elementary reactions for each pathway.

#### 2.3.1. Analysis of Formic Acid Formation Pathway on Hydrogen-Covered χ-Fe_5_C_2_ (510) Surface

The OH pathway and CO_2_ pathway are discussed first, and are shown in Figure 5. It is found that the Gibbs barrier energy of the first step of the elementary reaction in pathway 5 (yellow) is higher at 1.022 eV than that in pathway 6 (violet) at 0.744 eV. The former Gibbs reaction energy was 0.815 eV, whereas the latter Gibbs reaction energy was −0.160 eV; hence, the latter is more stable. Therefore, the first step of pathway 6, the elementary reaction O + H → OH, could occur more easily. In the second elementary reaction of pathway 6, CO + OH → COOH has a Gibbs barrier energy of 0.836 eV, but its effective Gibbs reaction energy is 0.588 eV, so the reverse reaction is more likely to occur. In contrast, Figure 5 shows that the effective Gibbs barrier energy of pathway 6 is 0.936 eV, which is determined by the last two steps of the radical reaction, while the effective Gibbs barrier energy of pathway 5 is 1.437 eV, and is much higher than that of pathway 6, so the effective Gibbs barrier energy of the OH pathway is defined as 0.936 eV.

For the CO_2_ pathway (e.g., reaction pathways 7 and 8 in Figure 5), the Gibbs barrier energy for the dissociated O with CO to produce CO_2_ is 0.853 eV, which is lower than that of the OH pathway. The Gibbs barrier energy in the second step of CO_2_ + H^1^ → COOH^1^ in reaction pathway 7 is 0.801 eV, while that of CO_2_ + H_2_ → COOH_2_ in reaction pathway 8 is 0.431 eV, which suggest that the O_2_ site of CO_2_ is more likely to undergo hydrogenation to produce COOH_2_. Thus, globally, the effective Gibbs barrier energy of reaction pathway 7 is 1.080 eV, determined by the first and second steps of the radical reaction, and that of reaction pathway 8 is 0.853 eV, determined by the first step of the radical reaction. It seems that the CO_2_ pathway is preferred, as its effective Gibbs barrier energy is 0.853 eV. However, the CO_2_ pathway is impossible, as the adsorption energy of CO_2_ is only 0.203 eV, which is much lower than the Gibbs barrier energy of CO_2_ + H_2_ → COOH_2_ (0.431 eV). After the formation of adsorption CO_2_, it is desorbed from the surface rather than undergoing the hydrogenation reaction.

The CO pathway is shown in Figure 6. The Gibbs barrier energy for the first-step elementary reaction CO + HO → COH in reaction pathways 1 and 2 is 1.788 eV; hence, its effective Gibbs barrier energy is theoretically ≥1.788 eV, and these two reaction pathways do not occur easily. As for reaction pathways 3 and 4, the Gibbs barrier energy of the first-step elementary reaction CO + H^C^ → CHO is 1.022 eV, and its Gibbs reaction energy is 0.815 eV, so the generation of CHO is unfavorable, which means that the occurrence of this reaction pathway is difficult. From Figure 6, it can be seen that the effective Gibbs barrier energies of reaction pathways 3 and 4 are 1.542 eV and 1.710 eV, respectively, and the effective Gibbs barrier energy of reaction pathways 1 and 2 is theoretically ≥1.788 eV. Therefore, the effective Gibbs barrier energy of the CO pathway is 1.542 eV, which is much higher than that of the OH pathway (0.936 eV) and the CO_2_ pathway (0.853 eV). It is very difficult for the CO pathway to occur.

#### 2.3.2. Charge Analysis of the Rate-Control Step in the Formation of Formic Acid on a Hydrogen-Covered χ-Fe_5_C_2_(510) Surface

The above analyses show that the effective Gibbs barrier energy for the CO hydrogenation reaction to form CHO (1.022 eV) and COH (1.788 eV) are high. In comparison, the effective Gibbs reaction barrier energy of O + CO → CO_2_ and OH + CO → COOH are only 0.853 eV and 0.836 eV, respectively. These four elementary reactions are the rate-controlled steps of their respective pathways, and their Gibbs barrier energies are very different from each other. Bader charge analysis is carried out to investigate the reasons in detail. The results are given in Table 2.

All these four elementary reactions take place all on the C reaction sites in CO; hence, the charges at the C reaction site of each elementary reaction transition state would be important and are shown in Table 2: elementary reaction 2 (3.665) > elementary reaction 1 (3.423) > elementary reaction 3 (3.092) > elementary reaction 4 (3.014). This result is consistent with the magnitude of the corresponding Gibbs barrier energy and suggests that the high Gibbs barrier energy for these four elementary reactions is mainly determined by the charge density of the C reaction site. The greater the charge density of the C site during the reaction, the more difficult it is for other species to react with CO. Thus, CO in the vicinity of dissociated O is more likely to react with oxygen atoms to form CO_2_ or with OH to form COOH.

#### 2.3.3. Summary

The above analyses show that formic acid is difficult to produce via the CO and CO_2_ pathways, and the optimal pathways of the three pathways (OH, CO_2_, and CO pathways) for the removal of dissociated oxygen in the form of formic acid are summarized and comparatively analyzed in Figure 7. It is found that the effective Gibbs barrier energy for the formation of formic acid via the CO pathway (green) is 0.936 eV, which is much lower compared to that of the CO pathway (red) of 1.437 eV. In addition, formic acid cannot be produced via the CO_2_ pathway (black 0.853 eV) either, because CO_2_ desorbs very easily. Therefore, CO + OH → COOH + H → HCOOH is the optimal pathway for the formation of formic acid when all pathways are compared.

In summary, the lowest effective Gibbs barrier energy for the removal of dissociated O in the form of formic acid is 0.936 eV, which is not much higher than that of the CO activation pathway (0.730 eV) [15]. Therefore, formic acid formation is kinetically feasible if the Fischer–Tropsch synthesis reaction can proceed. However, there are no literature reports of formic acid production in the Fischer–Tropsch synthesis reaction, which may be due to competition from other species. Hence, the reaction pathways for the removal of dissociated O in the form of CO_2_, H_2_O, and alcohols will be discussed in the next part to find the limiting factors for formic acid formation.

### 2.4. Other Represented Oxygenated Species Formation Pathways on the Hydrogen–Covered χ-Fe_5_C_2_(510) Surface and their Competition with Formic Acid

The possibility of formic acid formation was discussed and demonstrated in Section 2.3, but no formic acid formation has been reported in experiments, which may be because the formic acid formation pathway is unable to outcompete other species. Therefore, the O removal pathway with other oxygenated compounds (mainly CO_2_, H_2_O, alcohols, and aldehydes) are discussed, and the comparison between the formic acid formation pathway and other oxygenated compound formation pathways is investigated to identify the key factors limiting the formation of formic acid.

#### 2.4.1. The H_2_O and CO_2_ Formation Pathway on the Hydrogen-Covered χ-Fe_5_C_2_(510) Surface

CO_2_ and H_2_O are the main products of the removal of dissociated oxygen from the Fischer–Tropsch synthesis process and have been studied thoroughly [14,21]. However, the reaction pathways and effective Gibbs barrier energy for the removal of dissociated O in the form of H_2_O and CO_2_ are still considered based on the modelling and calculation accuracy of this work, so a comparison between the formic acid formation pathway and other oxygenated compound formation pathways is meaningful. The results are shown in Figure 8.

On the hydrogen-covered χ-Fe_5_C_2_(510) surface, the two-step hydrogenation reaction of dissociated O to produce H_2_O is mainly considered in the H_2_O formation pathway, the effective Gibbs barrier energy of which is 0.744 eV. Meanwhile, for the CO_2_ formation pathway, the CO + O (0.852 eV) route < CO + OH → COOH (1.342 eV) route < CO + H + O → COOH dehydrogenation (1.788 eV) route, indicating that if the dissociated O on the surface of χ-Fe_5_C_2_(510) is removed in the form of CO_2_, then the direct reaction of CO with dissociated O to produce CO_2_ is preferred. The Gibbs barrier energy of the reaction for the formation of CO_2_ (0.852 eV) is slightly higher than the Gibbs barrier energy for the formation of H_2_O (0.744 eV). Therefore, the dissociated O on the Fischer–Tropsch synthesis surface is preferentially removed as H_2_O, which is consistent with the results of Zhang’s [14] study and provides evidence that the calculations in this paper are credible.

#### 2.4.2. The Methanol Formation Pathway on a Hydrogen-Covered χ-Fe_5_C_2_(510) Surface

Monohydric alcohols are also one of the possible products of the O removal pathway, of which methanol is representative [51]. In the designed reaction network shown in Figure 4, the black main channel represents the stepwise hydrogenation of CO to methanol, where it can be divided into four levels in terms of product classification: 1, CHO and COH; 2, CHOH and CH_2_O; 3, CH_2_OH and CH_3_O; and 4, CH_3_OH. Figure 9 shows the Gibbs barrier energy for the stepwise hydrogenation of CO to produce methanol on hydrogen-covered χ-Fe_5_C_2_(510).

At level 1, the effective Gibbs barrier energy for the hydrogenation of CO to CHO (1.022 eV) is much lower than that for COH (1.788 eV). Therefore, CO is more likely to undergo hydrogenation at the C site to produce CHO species. Similarly, comparing the effective Gibbs barrier energy of each basic reaction at levels 2 and 3, it is found that the corresponding species also tend to undergo hydrogenation at the C site first, such as CHO + H → CH_2_O, CH_2_O + H → CH_3_O, with the effective Gibbs barrier energy of 0.251 eV and 0.221 eV, respectively. Finally, when the C site is saturated, only then does hydrogenation occur at the O site: CH_3_O + H → CH_3_OH.

In short, it is easy to see that the Gibbs barrier energy of the stepwise hydrogenation from CO to methanol is higher on the C-site than on the O-site hydrogenation at all levels. The Gibbs barrier energy of the O-site hydrogenation reaction decreases with the increase in the number of H atoms at the C-site (the effective reaction Gibbs barrier energy for the O-site hydrogenation at all levels: 1, 1. 7611 eV > 2, 1.275 eV > 3, 0.4835 eV > 4, 0.2739 eV). Thus, the main reaction pathway for the stepwise hydrogenation of CO to methanol is CO + H → CHO + H → CH_2_O + H → CH_3_O + H → CH_3_OH, with the effective Gibbs barrier energy of 1.412 eV, which is the highest effective Gibbs barrier energy of the generated species considered. In addition, the stepwise hydrogenation of CO to methanol involves the formation of formaldehyde (at level 2), of which the effective energy barrier is mainly determined by the first step CO + H → HCO of 1.022 eV.

#### 2.4.3. Competitive Analysis of the Formic acid Formation Pathway with the Formation Pathways of other Oxygenated Species on the Reaction of the Hydrogen-Covered χ-Fe_5_C_2_(510) Surface

After clarifying the preferred typical O removal product formation pathway, the Gibbs barrier energy of the formic acid formation pathway and each other product formation pathway can now be plotted in Figure 10.

Comparison of the effective Gibbs barrier energy of the optimal formation pathways for the different products in Figure 10, in order from smallest to largest: H_2_O (0.744 eV) < CO_2_ (0.853 eV) < HCOOH (0.936 eV) < CH_2_O (1.022 eV) < CH_3_OH (1.421 eV). The effective Gibbs barrier energy of the pathways for methanol and formaldehyde formation are much higher than those for the formation of other products, so that the production of alcohols and formaldehyde is kinetically unfavorable. In addition, it is seen in detail in Section 2.3.1. that it is also difficult to obtain formic acid via the CO pathway due to the fact that the effective reaction energy barrier of the CO pathway (1.542 eV) is much higher than that of the OH and CO_2_ pathways. It is found that there is a competitive relationship between CO + OH → COOH and H + OH → H_2_O, and the effective Gibbs barrier energy of the elementary reaction is COOH (0.836 eV) > H_2_O (0.553 eV). Because there are a large number of H atoms on the surface of χ-Fe_5_C_2_, which is favorable for the hydrogenation of OH to water, it is difficult for formic acid to be formed via the OH pathway reaction. In addition, it is discussed in Section 2.3 that the formation of formic acid via the CO_2_ pathway is difficult due to the low desorption energy of CO_2_. In summary, on the catalyst surface, the fact that formic acid is in competition with, and in a disadvantageous position with respect to, the pathways for the formation of H_2_O and CO_2_, results in the absence of formic acid formation.

What is more, the kMC simulation is applied to assist in finding the major reaction pathways, as only drawing energy profile diagrams and then determining reaction pathways cannot exhaust all the possible reaction pathways. Based on the kinetic data provided via DFT calculations, the kMC simulation could produce the reaction frequency of each elementary reaction, and the reaction pathways can be accurately obtained via the comparison of their reaction frequencies. 

The DFT results regarding the radical reactions considered are given in Appendix A, while the radical reactions considered in the kMC simulation are given in Appendix A; the simulated temperature was 600 K, the simulated pressure was 2 MPa, the ratio of CO to H_2_ in the feed composition was 1:1, and the simulation time was 1.0 × 10^−3^ s. When the kMC simulation system reached stability, the kMC product distribution and reaction frequency data were obtained, and the results are shown in Figure 9 (black arrows indicate that species undergo hydrogenation; red arrows indicate that species react with O; blue arrows indicate that species react with OH; green numbers indicate the net reaction frequency; and the absence of a number next to the arrow indicates that the reaction did not occur).

In the kMC simulation system (Figure 11), the reaction frequency of the formic acid, formaldehyde, and methanol formation pathway is 0; the reaction frequency of the elementary reaction H + OH → H_2_O is 3450, and the reaction frequency of the dissociated O removal in the form of CO_2_ is 1207, suggesting that H_2_O and CO_2_ are the main oxygen-containing compounds in the iron-based iron Fischer–Tropsch synthesis catalysts, and that the dissociated O is also more easily removed as H_2_O. While the primitive reaction CO + OH → COOH did not occur, surprisingly, there was still a very small amount (net reaction frequency = 22) of CO_2_ hydrogenation to produce COOH, but no progress in the reaction to produce HCOOH, which can be attributed to the increase in energy required due to the change in conformation of COOH.

#### 2.4.4. Summary

In summary, the combination use of DFT and kMC shows that the dissociated oxygen on χ-Fe_5_C_2_(510) is mainly removed in the form of CO_2_ and H_2_O, and the effective Gibbs barrier energy of the H_2_O production pathway is relatively lower, so more H_2_O is produced, while it is actually more difficult to produce HCOOH on χ-Fe_5_C_2_(510). The main reasons for the limitation of formic acid production are as follows: 1. If CO is hydrogenated first to form COH (1.788 eV) or CHO (1.022 eV), then it is difficult to form HCOOH due to the effective Gibbs barrier energy. 2. When CO reacts with O to form CO_2_, CO_2_ cannot be further hydrogenated to form HCOOH because it is easily desorbed. 3. When considering the OH pathway, because there are a large number of H atoms on the hydrogen-covered surface, OH is more likely to react with H atoms to form H_2_O, which also blocks the formation of HCOOH.

Based on the results of this work, two solutions are proposed to try to remove the dissociated O in the form of formic acid: firstly, increasing the adsorption energy of the catalyst for CO_2_ and the probability of the CO_2_ hydrogenation reaction occurring, by modifying the iron-based catalyst with other elements. Secondly, increasing the CO coverage on the catalyst surface so that CO could occupy more top-site iron atoms on the catalyst surface and reduce the occupancy of the hydrogen at the same time to prevent the hydrogenation reaction of OH species. The above analysis makes it possible to remove the dissociated O in the form of formic acid and, thus, achieve the resourceful use of dissociated O.

## 3. Conclusions

In this paper, the possibility of removing dissociated O by replacing H_2_O or CO_2_ (economically worthless byproducts) with HCOOH is proposed and discussed. First, the reaction mechanism of Fischer–Tropsch synthesis of formic acid was investigated via DFT simulation, and it was found that the best pathway to remove dissociated O in the form of formic acid is the OH pathway, through which OH + CO → COOH + H → HCOOH, with the effective Gibbs barrier energy of 0.936 eV. This result is close to the effective Gibbs barrier energy of 0.73 eV for the CO activation pathway, suggesting theoretical formic acid formation on the surface of iron-based Fischer–Tropsch synthesis catalysts. However, formic acid was not found in the experimental results, which could be attributed to the fact that the process used to form formic acid cannot compete with other species.

To prove this hypothesis, the removal of dissociated oxygen in the form of formic acid was compared with the formation pathways of CO_2_, H_2_O, and alcohol via DFT-kMC. Three main factors limiting the formation of formic acid were identified. 1. If the hydrogenation of CO is considered first, then the formation of HCOOH is hindered by the high effective Gibbs barrier energy for the formation of COH (1.788 eV) or CHO (1.022 eV). 2. If CO reacts first with O to form CO_2_, then it is difficult to hydrogenate it to HCOOH because CO_2_ is readily desorbed. 3. If the CO + OH pathway is considered, then OH reacts more readily with H atoms to form H_2_O than with CO to form COOH due to the hydrogen-covered effect.

There are two ways to attempt to remove dissociated oxygen in the form of formic acid, based on the reaction mechanism that limits the formation of formic acid. Firstly, the ability of the catalyst to adsorb carbon dioxide and the possibility of a carbon dioxide hydrogenation reaction can be improved by modifying the iron-based catalyst with other elements, which can be considered as the introduction of elements such as Cr, Mn, and Pd [52]. Secondly, by increasing the CO coverage on the catalyst surface while decreasing the hydrogen coverage, the likelihood of OH species undergoing a hydrogenation reaction is reduced. The introduction of the element Pb can also be considered, drawing on the idea of Pb electrocatalytic Fischer–Tropsch synthesis for the highly selective generation of formic acid [53].

## Figures and Tables

**Figure 1 molecules-28-08117-f001:**
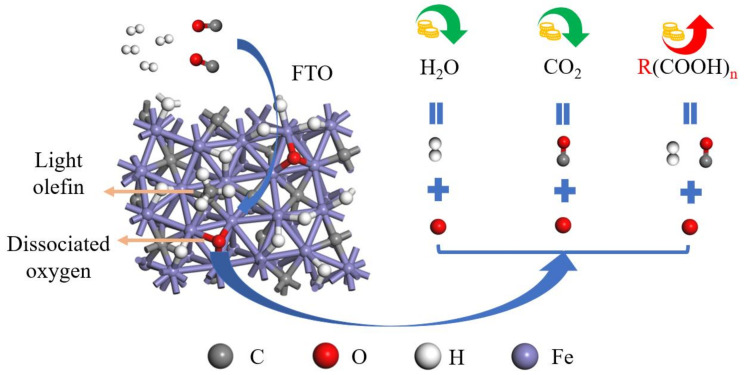
Possible forms of removal of dissociated oxygen in Fischer–Tropsch synthesis.

**Figure 2 molecules-28-08117-f002:**
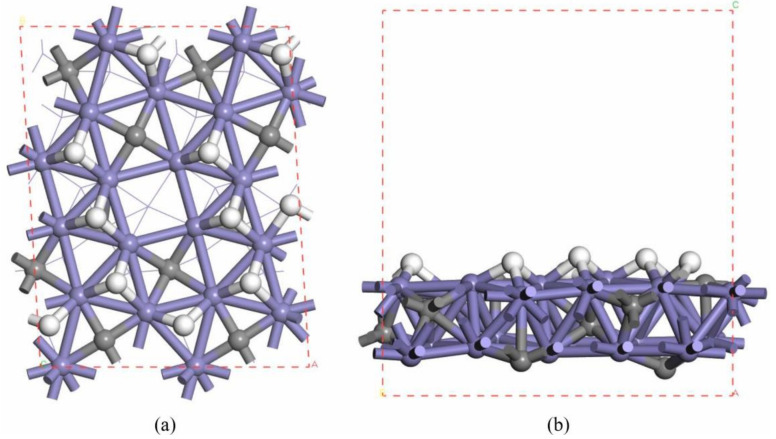
Model used for DFT computation (blue: iron atoms, grey: carbon atoms, white: hydrogen atoms). (**a**) Top view of χ-Fe_5_C_2_(510) with consideration of the hydrogen coverage effect; (**b**) side view of χ-Fe_5_C_2_(510) with consideration of the hydrogen coverage effect.

**Figure 3 molecules-28-08117-f003:**
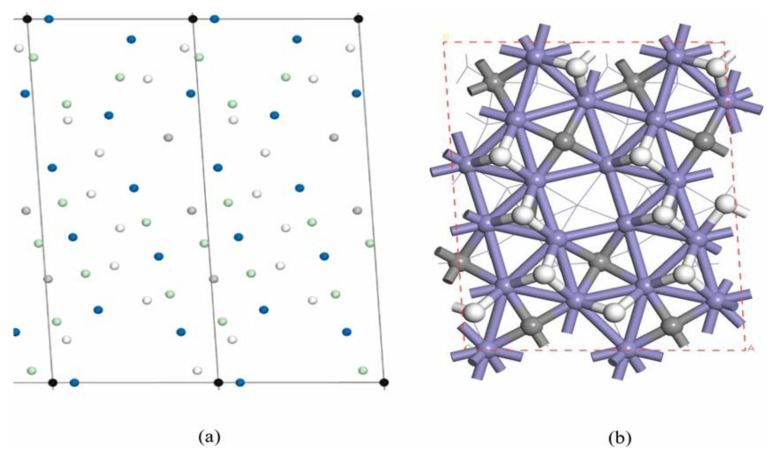
Model used in the DFT calculation based on the kMC method. (**a**) kMC model (blue: top sites of the iron atoms; green: bridge sites of the iron atoms; white: 3-fold sites of the iron atoms; gray: 4-fold sites of the iron atoms); (**b**) corresponding DFT model (blue: iron atoms; gray: carbon atoms; white: hydrogen atoms).

**Figure 4 molecules-28-08117-f004:**
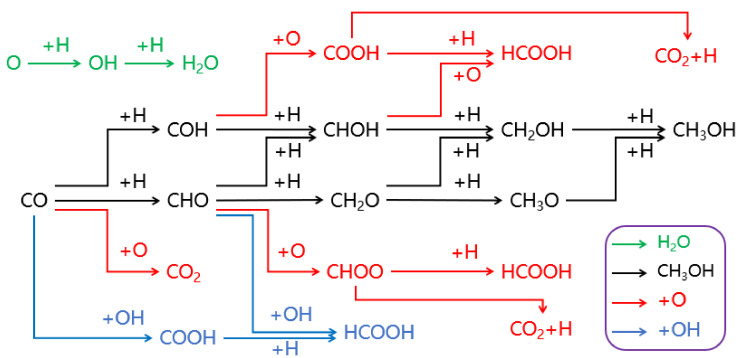
Possible dissociated O removal pathways for χ-Fe_5_C_2_(510) surfaces covered with hydrogen.

**Figure 5 molecules-28-08117-f005:**
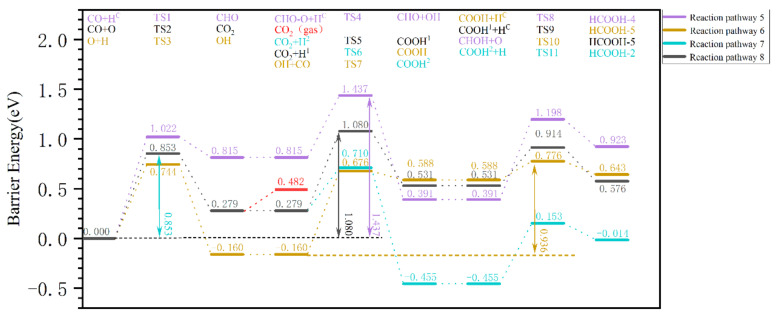
The effective Gibbs barrier energy diagram of the pathway for the removal of dissociated O from the surface of hydrogen-covered χ-Fe_5_C_2_(510) in the form of (OH and CO_2_ pathway) formic acid.

**Figure 6 molecules-28-08117-f006:**
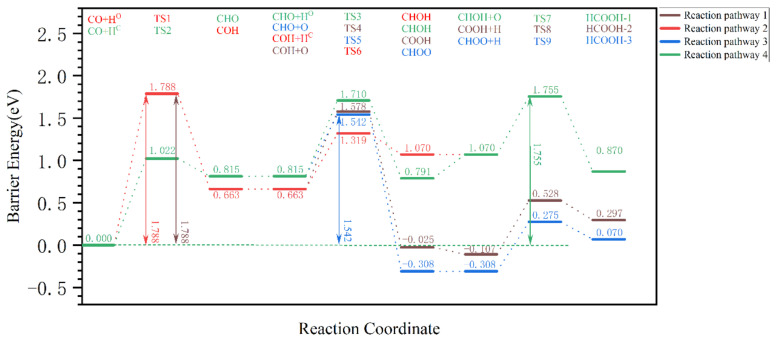
The effective Gibbs barrier energy diagram for the pathway of dissociated O removal from the hydrogen-covered χ-Fe_5_C_2_(510) surface in the form of (CO pathway) formic acid.

**Figure 7 molecules-28-08117-f007:**
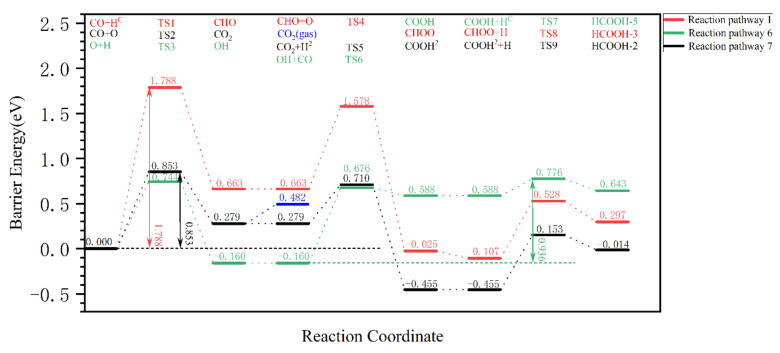
The effective Gibbs barrier energy diagram for the removal pathway of dissociated oxygen from the surface of hydrogen-covered χ-Fe_5_C_2_(510) in the form of (OH, CO_2_, CO pathway optimal pathway) formic acid.

**Figure 8 molecules-28-08117-f008:**
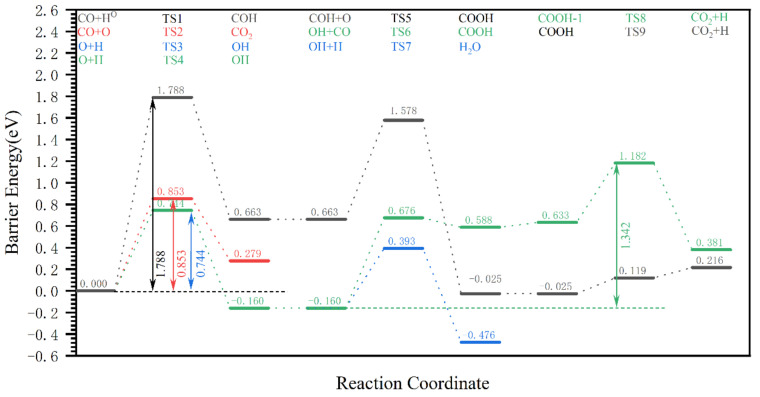
The Gibbs barrier energy diagram for the pathway of dissociated O removal as CO_2_/H_2_O on the hydrogen-covered χ-Fe_5_C_2_(510) surface.

**Figure 9 molecules-28-08117-f009:**
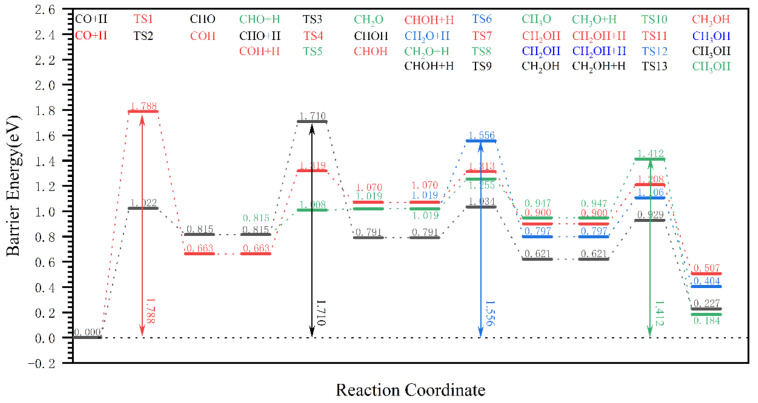
The effective Gibbs barrier energy diagram for the dissociated O removal pathway in the form of methanol on the hydrogen-covered χ-Fe_5_C_2_(510) surface.

**Figure 10 molecules-28-08117-f010:**
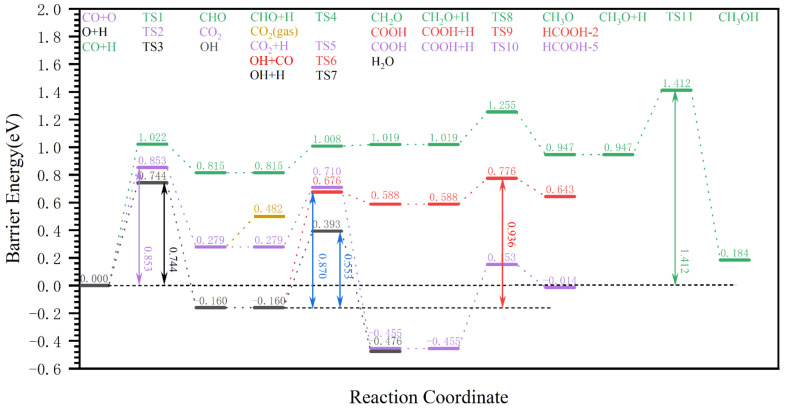
Comparison of the effective Gibbs barrier energy for the optimal removal pathways for each form of dissociated O from the hydrogen-covered χ-Fe_5_C_2_(510) surface.

**Figure 11 molecules-28-08117-f011:**
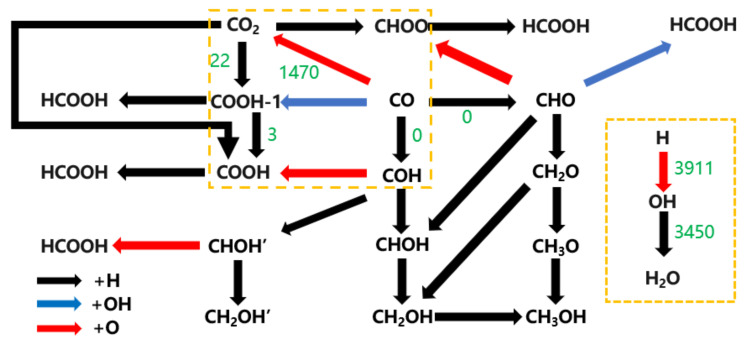
Schematic representation of the process of dissociated O removal from the hydrogen-covered χ-Fe5C2(510) surface.

**Table 1 molecules-28-08117-t001:** Detailed table of the elementary reactions for the reaction pathway of the hydrogen-covered surface reaction of χ-Fe_5_C_2_(510) to formic acid.

Reaction Pathway	Step 1: Elementary Reaction	Step 2: Elementary Reaction	Step 3: Elementary Reaction
1	CO Oxygen side hydrogenation CO + HO → COH	COH carbon site oxygenated COH + O → COOH	COOH carbon site hydrogenation COOH + H^C^ → HCOOH
2		COH carbon site hydrogenation COH + H^C^ → CHOH	CHOH carbon site oxygenated CHOH + O → HCOOH
3	CO carbon site hydrogenation CO + H^C^ → CHO	CHO carbon site hydrogenation CHO + O → CHOO	CHOO Oxygen side hydrogenation CHOO + HO → HCOOH
4		CHO Oxygen side hydrogenation CHO + HO → CHOH	CHOH carbon site oxygenated CHOH + O → HCOOH
5	CO carbon site hydrogenation CO + H^C^ → CHO	Dissociation O hydrogenation O + H → OH	CHO reacts with OH HCO + OH → HCOOH
6	Dissociation O hydrogenation O + H → OH	OH attacks the carbon site of CO OH + CO → COOH	COOH carbon site hydrogenation COOH + H^C^ → HCOOH
7	CO carbon site oxygenated CO + O → CO_2_	Hydrogenation of CO_2_ at the O_1_ Site CO_2_ + H → COOH	COOH carbon site hydrogenation COOH + H^C^ → HCOOH
8	CO carbon site oxygenated CO + O → CO_2_	Hydrogenation of CO_2_ at the O^2^ CO_2_ + H → COOH	COOH carbon site hydrogenation COOH + H^C^ → HCOOH

**Table 2 molecules-28-08117-t002:** Charge analyses of the rate-determining steps of the reaction pathway for the hydrogen-covered χ-Fe_5_C_2_(510) surface reaction to produce formic acid.

Elementary Reaction	Reaction Site	CHARGE		
		Initial State	Transition State	End State
1. CO + H → CHO	H	0.795	0.840	1.170
	C	3.493	3.422	3.147
	O	7.013	7.002	7.070
2. CO + H → COH	H	1.184	0.584	0.277
	C	3.126	3.665	3.757
	O	7.036	7.008	7.201
3. CO + O → CO_2_	Dissociated O	6.820	6.774	7.012
	C	3.120	3.092	2.542
	O	7.031	6.976	7.015
4. CO + OH → COOH	H	0.206	0.407	0.296
	C	3.196	3.014	2.774
	O^H^	7.292	7.011	7.193
	O^C^	7.035	7.026	7.080

## Data Availability

The data used to support the findings of this study are included within the article.

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
