# Peer review of "Mechanistic Study on the Possibility of Converting Dissociated Oxygen into Formic Acid on χ-Fe_5_C_2_(510) for Resource Recovery in Fischer–Tropsch Synthesis"

_molecules, 2023, doi:10.3390/molecules28248117_

Round 1

Reviewer 1 Report

Comments and Suggestions for Authors

Comments on the Quality of English Language

Author Response

This is an important topic of discussion in the world of Fischer-Tropsch synthesis. I really enjoyed reading this manuscript and it systematically investigates the potential routes for formic acid formation using DFT and KMC simulation on Fe5C2 (510). The manuscript was organized very well and mainly supports their conclusions with theoretical findings. However, I do have some comments which are mainly dealing with knowledge lag that must be addressed before this manuscript I believe suitable for publication in Molecules.

  1. Page 1, Line 14, Define DFT and kMC.

Our reply: We are thanks to your advice. We have changed the word "DFT and kMC" in the abstract to "Density Functional Theory (DFT) and kinetic Mont Carlo (kMC) "

  1. Did the authors also consider using Fe carbide other than x-Fe5C2 or oxide or oxyhydroxide phases of Fe to study formation of formic acid?

Our reply: Thank you for your valuable advice. The research of this paper is based on iron-based Fischer-Tropsch catalyst, which mainly contains many kinds of phases, such as iron oxide, iron carbide, metal iron and so on. Among these phases, the χ-Fe5C2 phase is considered to be the main active phase of the Fe-based Fischer-Tropsch synthesis catalyst, while the χ-Fe5C2(510) surface is the main exposed surface of the carbide phase (mentioned in the third paragraph of the preface). Hence, the main exposure surface of the main active phase catalyzed by Fe-based Fischer catalyst (χ-Fe5C2) was selected to study the factors limiting the formation of formic acid. If necessary, other phases will be considered to study the formation of formic acid in the later.

  1. Check this sentence: page 6, line 192 “The lattice gas model is the model used for the kMC simulations in this thesis.”

Our reply: We are sorry for our misuse of words. Here we have changed it to "The lattice gas model is the model used for the kMC simulations in this paper.".

  1. Check this sentence: page 7, line 217 “Finally, finding the possible surface species who which can react with dissociated O or OH in the main pathway to form formic acid are considered,”

Our reply: We apologize for making mistakes in grammar. We have deleted "who".

  1. Does these models (DFT & KMC) account the interference of other reaction intermediates other than species mentioned in Schemes in Table 1 for adsorption energy calculations? The surface coverage of H or CO is not just enough to make predictions on energy level of elementary reaction steps involved.

Our reply: Thank you for your useful advice. First, H and CO are the predominant coverages, and while the effect of derivatives from CO hydrogenation on the primitive reactions does exist, it is not a significant proportion, so we hold off on considering it. For the products involving chain growth, the main reaction site is the quaternary site of Fe, which is far away from the reaction site we studied, so we do not consider the effect of other species. The main influence still comes from the coverage of the catalyst surface with CO and H. The computational modeling in this paper is based on the hydrogen coverage effect, which you can see in the second paragraph of section 2.2, so for the surface coverage of H the surface coverage has been fully considered in this paper. For the surface coverage of CO, according to the current Fischer-Tropsch synthesis feed CO is generally less than H2, so there is indeed a lack of consideration here, but according to our calculations, to a certain extent, it can be a better explanation of the factors limiting the generation of formic acid. Based on the results of our calculations, in our subsequent studies we will also consider increasing the feed ratio of CO and increasing the pressure to adsorb more CO on the surface of the catalyst thus increasing the probability of the occurrence of the primitive reaction CO+OH=COOH and facilitating the formation of formic acid. Thank you again for your positive comments and valuable suggestions to improve the quality of our manuscript.

  1. Can authors comment on whether these predictions may be extrapolated to acetic acid as it is the next homologue in this series (formic acid, acetic acid, propionic acid, butanoic acid)? If the authors feel that it could differ from current observation, then what might be the cause and how do you deal with it in terms of mechanistic viewpoint.

Our reply: Thank you for providing meaningful thoughts. The conclusions for formic acid cannot be extrapolated to acetic acid, propionic acid, etc., because formic acid only involves the process of CO hydrogenation and oxygenation, but acetic acid, etc., also involves the process of carbon chain growth. First, to explain the reason why we have only investigated the possibility of formic acid formation by Fischer-Tropsch synthesis: as suggested at the end of the first paragraph of our preface, " Literature review revealed that O atoms are dissociated and left on the catalyst sur- face during Fischer-Tropsch synthesis and the majority of these dissociated O are removed from the surface in the form of H2O or CO2, which would reduce the atom economy. "In order to improve atom utilization, We analyzed in the second paragraph the species that can be used to remove dissociated oxygen by substituting H2O and CO2, which must form products in which one C atom can carry away two or more O atoms, and the products that satisfy the conditions are formic and polyacidic acids (acetic, propionic, etc. do not meet this requirement). Among them, formic acid, in addition to being non-toxic, high energy density, renewable and degradable, has great potential for green organic synthesis and biomass conversion, and has a wide range of applications in new energy utilization. So we chose formic acid as our target product. Secondly, if we consider acetic acid, propionic acid, butyric acid, then the whole reaction network will be very large, and the corresponding workload will be back growing. In view of the fact that, at present, the purpose of our research and the results of our research can, to a certain extent, better explain the factors limiting the formation of formic acid in Fischer-Tropsch synthesis and can provide ideas for its solution, we may not consider studying acetic acid, propionic acid, butyric acid, and so on, any more. Thirdly, according to the existing literature reports, the products of Fischer-Tropsch synthesis are a small amount of acetic acid, propionic acid, but no formic acid. Therefore, the conclusion of formic acid should be different from the formation mode of acetic acid. If we want to discuss this problem, we need to study again through calculation. The formation mode of acetic acid is much more complicated than formic acid, and all the reaction path strength of C2 needs to be considered. According to the existing calculation results, we speculate that the formation path of acetic acid may be: H3C may have a great influence on the combination of H3CCO and OH to form C-O bond.

  1. It is completely understandable that formic acid is difficult to form on iron carbide due to other competitive reactions such as CO2 and H2O formation which are thermodynamically more favorable. But what are the suggestions from the authors to the reader that the study could bring in. So that catalyst can be formulated such that one can utilize O more effectively in this regard. This must be elaborated in the summary section.

Our reply: Thank you for your valuable advice. In the last paragraph of the fourth section, we have put forward a solution based on the results of the calculation. combined with your opinions, we have refined the specific ideas to provide readers with a specific scheme for the preparation of catalysts. “Firstly, the ability of the catalyst to adsorb carbon dioxide and the possibility of carbon dioxide hydrogenation reaction can be improved by modifying the iron-based catalyst with other elements, which can be considered as the introduction of elements such as Cr, Mn, and Pd” and “The introduction of the element Pb can also be considered, drawing on the idea of Pb electrocatalytic Fischer-Tropsch synthesis for the highly selective generation of formic acid.”

  1. Page 11, line 310-312 “……lowest effective Gibbs barrier energy for removal of dissociated O in the form of formic acid is 0.936 eV, which is not much higher than that of CO activation pathway (0.730eV)”. It is curious to know, what the effect of reaction temperature might be?

Our reply: Thank you for your meaningful input. Regarding the effect of reaction temperature on the effective Gibbs energy barrier, we have mentioned it in Eq. 2 in Section 2.1. Firstly, the E-electron energies obtained from the DFT transition state calculations are combined with the frequency calculations to obtain the zero-point energy (ZPE) corrections, the standard molar vibrational internal energy contributions, and the standard molar vibrational entropy data, which are corrected by Eq. 2 to obtain the standard molar Gibbs free energy. The temperature correction involved here is mainly the TS term. We set T here to 600 K according to the actual experimental temperature of the Fischer-Tropsch synthesis, and the S data obtained from the calculations for the initial, transition, and final states do not differ much from each other, and the temperature-dependent actual corrections to the TΔS= (TSTS- TSIS) for the final effective Gibbs energy barrier are all about 0.1-0.2 eV, so we believe that the temperature does not have much effect on the effective Gibbs energy barrier.

  1. In literature it has been argued that olefins and oxygenates are considered as the primary products of iron FT synthesis. Methanol is known to be involved in the chain propagation and the extent of this reaction was observed to be high compared to ethanol. What is the authors opinion on secondary reaction of methanol involved on iron carbide which could have the impact on the KMC simulations described in Figure 11? Any comment.

Our reply: Thank you for providing meaningful thoughts. Methanol is obtained by CO hydrogenation, which we have considered in the kMC simulation in figure 11. So we have considered the step-by-step hydrogenation of CO to methanol (the reverse reaction is also considered), and the possibility of formic acid formation from the intermediates in this process is also considered, such as HCOH+O=HCOOH, but the calculated data show that this is not possible. Therefore, we think that it is very difficult to form formic acid through a series of reactions after the formation of methanol, so we prefer to produce formic acid by dissociating O.

Reviewer 2 Report

Comments and Suggestions for Authors

The manuscript discusses the possibility of removing dissociation O from the surface of Fischer-Tropsch synthesis catalyst in the form of formic acid so as to improve atomic economy. By studying the reaction path of hydrogen-covered χ-Fe5C2 surface reaction to formic acid, and analyzing the possibility of formic acid formation, the competitive relationship of removing dissociation O in formic acid form and in the form of H2O, CO2, methanol and formaldehyde was analyzed, and the factors restricting formic acid formation were obtained, and the solutions were put forward. The calculated experimental data and conclusions provided by this work are valuable and have attracted the attention of many researchers working in the field of reaction mechanism and catalyst modification of Fischer-Tropsch synthesis. Therefore, I this manuscript can be published in molecules after some minor revisions as below:

1. Section 3.2.1. Explain clearly whether the source of the effective reaction barrier formed by formic acid through the OH pathway is pathway 5 or pathway 6.

2. Section 3.3.3. In this paper, the effective reaction energy barrier of formaldehyde reaction pathway is proposed, but there is no corresponding analysis.

3. Section 3.3.3. The second paragraph explains the reason why OH pathway and CO2 pathway can not produce formic acid through comparative analysis, but CO pathway is not mentioned here.

4. Unify the significant digits of the full text into 3 or 2 digits, such as "7.070" or "7.07".

5. Section 3.3.1. Pay attention to superscript, for example: H2O.

6. Section 3.3.3. Spelling error: coldis changed to could, and it is recommended to modify the corresponding statement.

Author Response

The manuscript discusses the possibility of removing dissociation O from the surface of Fischer-Tropsch synthesis catalyst in the form of formic acid so as to improve atomic economy. By studying the reaction path of hydrogen-covered χ-Fe5C2 surface reaction to formic acid, and analyzing the possibility of formic acid formation, the competitive relationship of removing dissociation O in formic acid form and in the form of H2O, CO2, methanol and formaldehyde was analyzed, and the factors restricting formic acid formation were obtained, and the solutions were put forward. The calculated experimental data and conclusions provided by this work are valuable and have attracted the attention of many researchers working in the field of reaction mechanism and catalyst modification of Fischer-Tropsch synthesis. Therefore, I this manuscript can be published in molecules after some minor revisions as below:

  1. Section 3.2.1. Explain clearly whether the source of the effective reaction barrier formed by formic acid through the OH pathway is pathway 5 or pathway 6.

Our reply: Thank you for pointing out the error for us. The effective Gibbs barrier energy of path 6 is 0.936 e less than the effective Gibbs barrier energy of path 5 is 1.437 eV, so the effective Gibbs barrier energy of the OH path comes from with the reaction path 6. And corrected the error as follows:" while the effective Gibbs barrier energy of pathway 5 is 1.437 eV, and is much higher than that of pathway 6"。

  1. Section 3.3.3. In this paper, the effective reaction energy barrier of formaldehyde reaction pathway is proposed, but there is no corresponding analysis.

Our reply: Thank you for your valuable advice. Taking your suggestion into account, we have the analysis of the effective reaction energy barriers for formaldehyde added in Section 3.3.1 “In addition, the stepwise hydrogenation of CO to methanol involves the formation of formaldehyde (At level 2), whose effective energy barrier is mainly determined by the first step CO + H = HCO of 1.022 eV.”and introduced in the comparison in Section 3.3.3.

  1. Section 3.3.3. The second paragraph explains the reason why OH pathway and CO2pathway can not produce formic acid through comparative analysis, but CO pathway is not mentioned here.

Our reply: We have added this section in response to reviewer comments. The details are as follows: "In addition, it was analysed in detail in Section 3.2.1 that it is also difficult to react to formic acid via the CO pathway due to the fact that the effective reaction energy barrier of the CO pathway (1.542 eV) is much higher than that of the OH and CO2 path-ways.".

  1. Unify the significant digits of the full text into 3 or 2 digits, such as "7.070" or "7.07".

Our reply: We apologize for the oversight of our different formatting. We have replaced "7.07" with "7.070" in table 2.

  1. Section 3.3.1. Pay attention to superscript, for example: H2O.

Our reply: We apologize for writing “H2O” incorrectly, and we have also changed some of the formatting errors "OH" and "OC" in Table 2.

  1. Section 3.3.3. Spelling error: “cold”is changed to “could”, and it is recommended to modify the corresponding statement.

Our reply: We have made correction according to the Reviewer’s comments. Replace "cold" with "could".

Reviewer 3 Report

Comments and Suggestions for Authors

1. This work presents the feasibility of using a Fe-based catalyst to transform dissociated oxygen into formic acid. But according to their findings, the production of formic acid is only theoretically feasible and the reaction in different states on a Fe-based catalyst does not demonstrate the energy advantage of producing formic acid. They proposed that "increasing the adsorption energy of the catalyst for CO2 and the probability of the CO2 hydrogenation reaction occurring, by modifying the iron-based catalyst with other elements." is one strategy to remove dissociated oxygen in the form of formic acid. However, there is not enough articles reported that the addition of elements promotes the synthesis of formic acid. Consequently, it is encouraged to use promoters in order to investigate the process of formic acid synthesis. Besides, the following two articles are provided for reference. (1) Mechanistic Insight into Hydrocarbon Synthesis via CO2 Hydrogenation on χ‑Fe5C2 Catalysts; (2) Mechanistic Understanding of Hydrocarbon Formation from CO2 Hydrogenation over χ‑Fe5C2(111) and the Effect of H2O and Transition Metal Addition.

2. In this study, the ultimate state of several intermediary reaction steps is the same product, with a pretty similar adsorption energy on the surface. To strengthen the conclusion, it is advised to take into account the migration of the same species at various surface locations.

3. The coupling of CO and OH is said to have a relative advantage in the reaction pathways of each formic acid creation; however, this research does not go into additional detail regarding the formic acid's energy advantage reaction path. It is advised to expand research into the impact of surface reduction ability on OH production as well as the energy distribution of CO and OH reactions on surfaces with varying OH coverage.

4. There are some errors in the text, such as CH3OH in Figure 10 is written as CH4OH.

Comments on the Quality of English Language

no comments on English

Author Response

1.This work presents the feasibility of using a Fe-based catalyst to transform dissociated oxygen into formic acid. But according to their findings, the production of formic acid is only theoretically feasible and the reaction in different states on a Fe-based catalyst does not demonstrate the energy advantage of producing formic acid. They proposed that "increasing the adsorption energy of the catalyst for CO2 and the probability of the CO2 hydrogenation reaction occurring, by modifying the iron-based catalyst with other elements." is one strategy to remove dissociated oxygen in the form of formic acid. However, there is not enough articles reported that the addition of elements promotes the synthesis of formic acid. Consequently, it is encouraged to use promoters in order to investigate the process of formic acid synthesis. Besides, the following two articles are provided for reference. (1) Mechanistic Insight into Hydrocarbon Synthesis via CO2 Hydrogenation on χ‑Fe5C2 Catalysts; (2) Mechanistic Understanding of Hydrocarbon Formation from CO2 Hydrogenation over χ‑Fe5C2(111) and the Effect of H2O and Transition Metal Addition.

Our reply: Thank you for providing us with reference information to make the manuscript more informative. We first investigated the optimal path for the generation of possible formic acid on the surface of χ-Fe5C2(510), and later, by studying the competing relationship between the generation of formic acid and CO2, H2O, and methanol, we finally obtained the factors limiting the generation of formic acid, and targeted the solution, but it was not refined enough. Thank you for refining our thinking. We have added in the first paragraph of the Introduction " There are two ways to solve this problem: one is to promote further reaction of oxy-gen-containing compounds in the product to form valuable products, such as hydrogenation of CO2 to hydrocarbons[20]; the other is to find a product with high economic benefits that can replace CO2 and H2O. The former improves the economic benefit to a certain extent. However, it does not solve the problem of using the oxygen atoms, so it is better to find a substitute." and the last paragraph of the Conclusion added “Firstly, the ability of the catalyst to adsorb carbon dioxide and the possibility of carbon dioxide hydrogenation reaction can be improved by modifying the iron-based catalyst with other elements, which can be considered as the introduction of elements such as Cr, Mn, and Pd[52].”

  1. In this study, the ultimate state of several intermediary reaction steps is the same product, with a pretty similar adsorption energy on the surface. To strengthen the conclusion, it is advised to take into account the migration of the same species at various surface locations.

Our reply: Thank you for your meaningful advice. In the multi-scale simulation, it is very necessary to consider the process of mutual transformation of intermediate species, we actually consider the process of mutual transformation of intermediate species, but it is not reflected in the article, which is our negligence. For this reason, we use HCOH as an example in the support information. Therefore, we used HCOH in the Supporting Information as an example to demonstrate the interconversion of different structures of the same species. In our calculations, there are three structures, HCOH, HCOH1, and HCOH2, respectively. In Figure S2, we show the structures of the initial, transition, and final states of HCOH transforming first to HCOH1, and then HCOH1 transforming to HCOH2, and HCOH can be regarded as a bridge between HCOH and HCOH2, respectively. The specific contents are as follows:

An example of mutual transformation of different adsorption structures of the same species.

In Figure S2, the interconversion of HCOH, HCOH1, and HCOH2 is shown as an example of the interconversion of different adsorption structures of HCOH.

HCOH           TS:HCOH→HCOH1                  HCOH1

HCOH1         TS:HCOH1→HCOH2             HCOH2

Figure S2 Interconversion processes of different adsorption structures of HCOH.

It is true that some of the same species do not consider mutual transformation, such as HCOOH, mainly because the species involved are difficult to transform into each other, but the actual desorption energy of HCOOH is only 0.24eV, so we think that it will be desorbed directly when HCOOH is generated.

  1. The coupling of CO and OH is said to have a relative advantage in the reaction pathways of each formic acid creation; however, this research does not go into additional detail regarding the formic acid's energy advantage reaction path. It is advised to expand research into the impact of surface reduction ability on OH production as well as the energy distribution of CO and OH reactions on surfaces with varying OH coverage.

Our reply: Thank you for providing meaningful thoughts. The reason why we do not consider the effect of different OH coverage is that if we increase the OH coverage on the surface, we need to increase the amount of dissociated oxygen on the catalyst surface. This means that the catalyst will be oxidized and deactivated, which is not what we want. We need the catalyst to maintain high activity, so the amount of O (OH) considered should be kept at a low level. It is indeed a very good idea to increase the coverage of CO on the surface of the catalyst. In the conclusion of the manuscript, we proposed to increase the coverage of CO on the surface of the catalyst to promote the formation of formic acid (in practice, it is considered to increase the feed ratio of CO and increase the reaction pressure), which will be reflected in our follow-up study to solve the formation of HCOOH.

  1. There are some errors in the text, such as “CH3OH” in Figure 10 is written as “CH4OH”.

Our reply: I'm sorry for our mistake. We have corrected the mistakes in Figure 10.

Round 2

Reviewer 3 Report

Comments and Suggestions for Authors

Accept in present form

Author Response

Thank you for your recognition